# Development of a Highly Sensitive Technique for Capturing Renal Cell Cancer Circulating Tumor Cells

**DOI:** 10.3390/diagnostics9030096

**Published:** 2019-08-14

**Authors:** Michio Naoe, Chiho Kusaka, Mika Ohta, Yuki Hasebe, Tsutomu Unoki, Hideaki Shimoyama, Takehiko Nakasato, Kazuhiko Oshinomi, Jun Morita, Kohzo Fuji, Yoshio Ogawa, Mana Tsukada, Masataka Sunagawa, Hikaru Ishii

**Affiliations:** 1Department of Urology, Showa University, Tokyo 142-8666, Japan; 2Department of Physiology, School of Medicine, Showa University, Tokyo 142-8666, Japan; 3Ishii-clinic Kyobashi Edogrand, Tokyo 104-0031, Japan

**Keywords:** circulating tumor cells, renal cell carcinoma, G250 antigen

## Abstract

Purpose: Liquid biopsy is becoming increasingly important as a guide for selecting new drugs and determining their efficacy. In urological cancer, serum markers for renal cell and urothelial cancers has made the development of liquid biopsy for these cancers strongly desirable. Liquid biopsy is less invasive than conventional tissue biopsy is, enabling frequent biopsies and, therefore, is considered effective for monitoring the treatment course. Circulating tumor cells (CTCs) are a representative liquid biopsy specimen. In the present study, we focused on developing our novel technology for capturing renal cell cancer (RCC)-CTCs using an anti-G250 antibody combined with new devices. Basic experiments of our technology showed that it was possible to detect RCC-CTC with a fairly high accuracy of about 95%. Also, RCC-CTC was identified in the peripheral blood of actual RCC patients. Additionally, during the treatment course of the RCC patient, change in the number of RCC-CTC was confirmed in one case. We believe that the technology we developed will be useful for determining the treatment efficacy and drug selection for the treatment of renal cell cancer (RCC). In order to solve issues such as thresholds setting of this technology, large-scale clinical trials are expected.

## 1. Introduction

Currently, the standard method for detecting cancer metastasis is biopsy. However, biopsy has the disadvantages of being time-consuming, a painful burden on the patient, and expensive.

Liquid biopsy using body fluids such as blood is attracting attention as a method of detecting metastasis instead of conventional biopsy. It is known that a few cells derived from cancer tissue, called circulating tumor cells (CTCs), infiltrate the blood and lymph fluid and circulate in the body of cancer patients. CTCs have been the subject of research and development worldwide, recently.

CTCs are suggested to be associated with metastasis, and are expected to be a useful liquid biopsy specimen that can be detected in the blood. Examining CTCs is expected to possibly enable the early detection of micro cancer that cannot be confirmed by image inspection. Also, it is possible to evaluate the therapeutic effect using a simple method much more quickly than conventional diagnostic imaging methods such as computed tomography (CT). Blood tumor markers are widely used to determine therapeutic effects and predict prognosis in various solid cancers.

However, in urological cancer treatment, there are no blood markers for renal cell cancer (RCC) and urothelial cancers including bladder, ureter, and renal pelvic cancers. On the other hand, drug therapy for RCC is diverse, including various molecular-targeted drugs and immune-oncology drugs, and indicators are necessary for drug selection and conversion. Indicators are also required to determine the therapeutic efficacy of these drugs.

Over the past 10 years, many research studies worldwide have been attempting to identify CTC in numerous cancer types. Currently, the CellSearch System^®^ (Veridex, USA) [1] is the only method of CTC identification approved by the US Food and Drug Administration (FDA). However, a recent challenge is the existence of CTCs that cannot be identified using the CellSearch System^®^. In this study, we demonstrate the superiority of our developed CTC detection technique over those previously available.

## 2. Materials and Methods

### 2.1. Stainability of Anti-G250 Antibody in Various Cancer Cell Lines

First, the stainability of renal cancer cell lines by the anti-G250 antibody was determined. RCC OS-RC-2 and VMRC-RCW cell lines were stained with an anti-G250 antibody (anti-carbonic anhydrase 9-PE human, Miltenyi Biotec). As a control, prostate (DU145 and LNCap) and bladder (T24 and KK47) cancer cell lines were also stained. Then, flow-cytometric analysis using the BD FACSCalibur™ was performed.

### 2.2. Concentration and Analysis of CTC (Using Celsee^®^ Combined with On-chip Sort^®^)

To collect and detect rare CTCs, they need to be concentrated by eliminating peripheral blood mononuclear cells (PBMCs) from whole blood samples first. Concentration of CTC was achieved using CelSee^®^ as mentioned above, which is a physical method independent of antigen–antibody reactions. Subsequently, On-chip Sort^®^ is used to analyze specimens concentrated using CelSee^®^ in which most PBMCs have been removed by filtration. Among the cells present in the blood, CD45-negative/G250-positive cells were defined and counted as RCC CTCs.

### 2.3. Cell Retrieval Using Celsee PREP100^®^ Combined with On-Chip Sort^®^

A different number of VMRC-RCW cells (0,10, 20, and 40) was spiked into 4 mL whole blood, which was then hemolyzed by adding RBC lysis buffer (10×, BioLegend, San Diego, CA, USA) and standing the mixture for 15 min at room temperature. Following centrifugation at 326× *g* for 5 min at room temperature, the supernatants were discarded, and the isolated cells were resuspended in 8 mL T buffer (On-chip Biotechnologies, Tokyo, Japan).

The settled blood was collected and resuspended in 2 mL phosphate-buffered saline (PBS) using the Celsee PREP100^®^ instrument (Celsee Diagnostics) following the protocol provided by the manufacturer.

Captured cells were then collected by reverse injection (Figure 1) and concentrated in 10–50 µL by centrifugation at 500× *g* for 10 min. Then, the cells were triple immunostained with anti-CD45 (PerCP anti-human CD45 antibody BioLegend), anti-epithelial cellular adhesion molecule (EpCAM), (PE anti-EpCAM (EBA-1, BD Biosciences), and anti-G250 antibodies, followed by flow cytometric analysis using the On-chip Sort^®^ to count the CTCs. As a first step, cells other than blood cells are recognized as anti-CD45-Ab negative fractions.

Then, within the anti-CD45-Ab negative fractions, cells positive for anti-G250-Ab or anti-EpCAM-Ab are recognized as RCC CTC. However, the sensitivity of these two markers for RCC CTC, sensitivity of anti-G250-Ab is much higher than that of anti-EpCAM-Ab. This is an epoch-making and excellent point compared to the conventional EpCAM-based CTC detection.

### 2.4. Patient Samples

This study was approved by the ethics committee of Ishii-clinic Kyobashi Edogrand in December 2017. Written informed consent was obtained from all study participants.

Peripheral blood samples (10 mL) were collected from patients into Cell-Free DNA BCT CE tubes^®^ (Streck) and the CTCs were identified within 24 h after collection. Furthermore, 4 mL of each patient’s blood was used for each CTC measurement.

## 3. Results

### 3.1. Anti-G250 Antibody Staining Pattern in Various Cancer Cell Lines

The stainability of the anti-G250 antibody was confirmed in various cancer cell lines. As shown in Figure 2, RCC cell lines showed high stainability for anti-G250 antigen, while other cell lines showed no stainability.

### 3.2. Identification Rate of RCC CTC Using On-Chip Sort^®^ with G250 Antibody

The On-chip Sort^®^ was used to identify RCC CTCs in 4 mL whole blood spiked with 50 or 100 RCC cells. The PBMCs and CTCs were distinguished by triple-staining with anti-D45, anti-G250, and anti-EpCAM antibodies. Furthermore, anti-EpCAM staining was performed to compare its stainability of RCC CTC with that of the anti-G250 antibody.

First, the anti-CD45 negative fraction was extracted and then the staining properties of anti-EpCAM and anti-G250 antibodies in anti-CD45-negative cells were evaluated. The results showed that in the sample containing 50 cells in 4 mL of whole blood, six cells were missed as CTCs when anti-EpCAM cells were defined as CTCs. Conversely, when anti-G250 antibody-positive cells were defined as CTCs, 38 of the 50 (76%) cells were identified. Similarly, in samples in which 100 cells were mixed with 4 mL whole blood, 75 cells (75%) were detected when cells showing positivity for anti-G250 antibody were detected (Figure 3). Based on the result of this experiment, we decided to ignore the stainability for anti-EpCAM antibody and defined anti-G250 antibody-positive and -negative cells as RCC CTCs.

### 3.3. RCC Cells Concentration Using CelSee^®^ and Spiked RCC Cells Counting Using On-Chip Sort^®^ with Anti-G250 Antibody

Based on the results of the CTC identification rate experiment, we next enriched CTCs using CelSee^®^ combined with the discrimination of these cells from PBMCs using On-chip Sort^®^. The accuracy rate of RCC CTC detection was very high at approximately 95% (Figure 4).

### 3.4. Pilot CTC Examination in Patients with Metastatic RCC

Patient characteristics and the results of the CTC number analysis are listed in Table 1. Thirteen patients were included in this study and as shown in Table 1, not all patients showed positivity in the CTC test. CTCs were not detected in cases where there was no hematogenous distant metastasis such as lung metastasis, no matter how large the tumor was, such as in Case No. 10.

Therefore, these results indicate that the CTC examination reflected the state of hematogenous metastasis.

In addition, CTC was detected in six cases of patients with a pathological diagnosis of ccRCC and, interestingly, also in one with papillary carcinoma (Case No. 2), as well as two of five cases with unknown histological type (Cases No. 7 and 9). Although, the pathological diagnoses were unknown in these cases, they may have a high probability of ccRCC. In Case No. 4, a change in the number of CTCs was observed during treatment. The patient originally had lung metastases, and a lung metastasectomy was performed, followed by administration of sunitinib, which was subsequently discontinued because of the patient’s finances. Then, the patient underwent a CTC test, after consenting. As shown in Figure 5, 10 CTCs/4 mL of blood were observed. 

A lung computed tomography (CT) scan was performed simultaneously with the CTC test and confirmed new lung metastasis. Therefore, the patient was advised to resume sunitinib, and complied. The CTCs disappeared approximately two weeks after sunitinib was initiated. However, since the CT scan showed a slight increase in lung metastasis, one CTC/4 mL of blood was observed when the CTC test was repeated. In the near future, we will be able to change the medication for this patient and this case is an example where the CTC test was useful for drug selection during treatment.

## 4. Discussion

### 4.1. History of CTC Identification

In the past decade, various CTC capturing methods have been explored and many, which were based on polymerase chain reaction (PCR), were problematic in their sensitivity and reproducibility. One of the many CTC capturing methods developed after the PCR-based methods, includes the CellSearch System^®^ established by Cristofanilli et al. [1], using automated immunostaining. This method is based on direct observation of CTCs under a fluorescent microscope and was anticipated to be a new cancer biomarker.

The CellSearch System^®^ was approved by the US Food and Drug Administration (FDA) for the prediction of progression-free survival (PFS) and overall survival (OS) in metastatic breast cancer in 2004. Its approval was further extended to monitoring treatment effect of metastatic breast cancer in 2006, and then prediction of PFS and OS in metastatic colon and prostate cancers in 2007 and 2008, respectively. However, as will be discussed below, several challenges have been reported since then.

Here, we discuss the principles of CTC identification used in the CellSearch System^®^ and their problems. Only several to several tens of CTCs are present in 1 mL of blood, which contains approximately 5 billion cells, which are mostly red blood cells and PBMCs. Consequently, capturing and isolation of CTCs are extremely difficult.

### 4.2. Regarding the Weak Point of CellSearch System^®^

The CellSearch System^®^, which was established by Cristofanilli et al. [2] using automated immunostaining based on direct observation of CTCs under a fluorescent microscope, was anticipated to be a new cancer biomarker. The CellSearch System^®^ obtained FDA approval for predicting the prognosis of patients with metastatic breast and prostate cancers. Although CellSearch^®^ is the only FDA-approved method, its basic principle involves identifying CTCs as epithelial cellular adhesion molecule (EpCAM)-positive cells and it is difficult to identify EpCAM-negative CTCs.

Epithelial-mesenchymal transition (EMT) is the ability of cells to migrate and penetrate other tissues, losing their shape as epithelial cells and cell adhesion function to surrounding cells when cancer cells invade or metastasize. EMT is thought to be closely involved in the most important aspects of cancer treatment. Cancer is originally characterized by epithelial cells. However, in highly malignant cancers, EMT often occurs with the loss of epithelial cell characteristics. Furthermore, although EpCAM is an antigen expressed on epithelial cell surface, its expression is known to be attenuated in cancer cells, which causes EMT, leading to the inability to identify CTCs using CellSearch^®^.

Additionally, EpCAM has been reported to be an epithelial cell-specific marker, which is highly expressed in breast, prostate, and colon cancers but not RCC [3,4,5].

The following sections describe the principle and problems of CTC identification, and the latest methods for identifying CTCs of RCC.

### 4.3. Microfluidic Chip of CelSee^®^

As an alternative to CellSearch^®^, new CTC detection methods are being developed sequentially, including a method using a microfluidic chip (microfluidic device method), which is thought to have the highest detection sensitivity. CelSee^®^ is a microfluidic device method that was awarded “The Scientist’s Annual Top 10 Innovations of 2015.” Compared with CellSearch^®^, which has been accredited by the US FDA, Celsee^®^ has been reported to have a high CTC capture rate. The basic principle of CelSee^®^ is explained as follows.

The microfluidic chip method was designed based on the principle that the deformability and diameter of CTCs and peripheral blood mononuclear cells (PBMCs), as white blood cells (WBCs), differs greatly. Consequently, CTCs can be captured by the device, whereas PBMCs can pass through because their characteristic deformation is high even when they are larger than the pore channel, whereas that of CTCs is less, precluding permeation. CelSee^®^ is an apparatus used for the process of concentration of CTCs [6]. The procedure involves placing a microfluidic chip on one manifold and the blood is allowed to flow through. Cells are subsequently capture in approximately 50,000 trapping chambers of the Microfluidic Chip.

### 4.4. On-chip Sort^®^

The cell sorter is a widely used device for detecting and collecting target cells, but its use is associated with some problems in detecting and collecting rare cells such as CTCs. For example, in a typical capillary type cell sorter, it is impossible to analyze the entire sample as a “dead volume” exists in the flow path, and contamination between samples is a concern because the same flow path is used. The rare cell-sorting method using the microchip type cell sorter On-Chip Sort^®^ (On-chip Biotechnologies, Tokyo, Japan) substantially addresses these problems. On-chip Sort^®^ is a novel benchtop cell sorter equipped with a disposable microfluidic device, allowing the detection and isolation of rare tumor cells for subsequent molecular analyses.

The advantages of On-chip Sort^®^ are as follows. (1) It adopts a method of extruding the applied sample by air, making it is possible to analyze the whole sample in the chamber. (2) The flow path length is at the micrometer level and, so, the dead volume is ≤ 0.01 μL. (3) The flow path system is completely in the exchange type microchannel chip and, therefore, no contamination occurs between samples. (4) The sample flow occurs in the microchannel chip. (5) The compactness of the device allows it to be placed in a safety cabinet, and it can be sterilized using a sterilized chip. All these features make cell sorting is possible [7].

As described above, when the EpCAM antigen is targeted, there are CTCs that cannot be identified and under extreme conditions, there is no antigen specific for all cancers. Since our goal is to identify renal CTCs in RCC, we decided to use a renal cancer-specific antibody that is specialized for identifying renal cancer cells. One such candidate antibody is the anti-G250 antibody.

### 4.5. G250 antigen

Because the expression of EpCAM antigen on RCC cell is low, other biomarkers have been explored for the detection of RCC-CTC. Although other biomarkers including p53, p21, hypoxia-inducing factor (HIF)-1α, caveolin-1 [8], and survivin have been reported as potential prognostic biomarkers for RCC patients, they are not located on the cell membrane. Therefore, improving the efficiency of CTC capture in RCC patients by developing alternative cell surface biomarkers remains a challenge [9]. Monoclonal antibody G250 (mAbG250) was isolated more than 25 years ago from a hybridoma produced from splenocytes of a mouse immunized with fresh human RCC cells [10].

Subsequently, the cancer-associated antigen G250 antigen (MN/CA9) was reported in detail for the first time by Oosterwijk [11].

Cancer-associated antigen G250 is glycoprotein present in cell membranes and nuclei and is considered a carbonic anhydrase isoenzyme. G250 antigen is expressed by virtually all ccRCC cells, but its expression in normal tissues is restricted. In addition, the most prominent known subtype of RCC is ccRCC at 70%. The mouse monoclonal antibody against human renal cell carcinoma mAb G250 specifically recognizes the ccRCC membrane antigen (G250).

G250 antigen has been shown to be expressed in 95% and 75% of primary tumors and metastatic lesions in immunohistological research, and its expression is hardly recognized in other normal tissues including the kidney. It is correct to describe the G250 antigen as a specific antigen of ccRCC [12,13,14]. As shown in Figure 2, RCC cells showed strong dyeability to the anti-G250 antibody, which has also been reported by other studies.

Therefore, we decided to use G250 antigen as a target in the detection of CTC instead of EpCAM antigen.

Bluemke et al. reported that the patient CTC count was an independent prognostic factor that correlated with lymph node invasion in RCC [15]. However, CTC research in RCC has shown little progress, which is largely due to the lack of appropriate surface markers that can be used to capture antigens [16]. Again, EpCAM is the major molecule used as the capture antigen for CTC research studies reported in the literature [17]. EpCAM is an epithelial cell-specific marker, which is highly expressed in breast, prostate, and colon cancers but not RCC [3,4]. Therefore, there is a need for another specific surface markers for capturing RCC-CTC to replace the EpCAM antigen.

One such candidate is G250 antigen. In this study, we showed, for the first time, the specificity of G250 to RCC. After CTC concentration using Celsee, we successfully discriminated between CTCs and PBMCs with high sensitivity using a combination of anti-CD45 and anti-G250 antibodies [18].

The capture rate of RCC-CTC using our technology exceeded 90%. Although a small number was involved in the clinical investigation, we propose that the CTC test using patient samples indicates the appropriateness of using this test on patients with hematological distant metastasis such as that of the lung, liver, and bone. We also presented cases where this CTC test was positive and CTC disappeared in subsequent drug treatment. In such cases, this test would be very useful in determining the usefulness of the current drug therapy.

Furthermore, if the number of CTCs does not decrease after a certain drug treatment is commenced, it may suggest the drug needs to be changed. Presently, there are approximately 100 CTC identification methods worldwide. Many of them are based on capturing of EpCAM-positive cells. However as mentioned above, those methods are inaccurate. While limited only to RCC, we believe our technology is extremely accurate compared to previous techniques. Currently, clinical research using our technology is about to begin. If the clinical research proves the usefulness of our technology, we believe that it will contribute greatly to the treatment of kidney cancer without biomarkers, although there are various drug options. Because the clinical trial we conducted included a very small population, further studies with larger a number of cases are needed to determine parameters such as the cut-off value.

## 5. Conclusions

Currently, research groups around the world are developing new CTCs detection and analysis technologies, which are reported every year. On-chip Sort is a unique cell sorter and CelSee^®^ is a revolutionary CTC concentrator that is not based on the conventional antigen-antibody method.

We developed a novel technique for the identification of RCC CTCs using CelSee^®^ combined with On-chip Sort^®^ using the G250 antigen, which is a crucial combination for identifying RCC CTCs with high accuracy. Henceforth, liquid biopsies using CTCs are expected to be used for a wide range of applications such as drug biomarker testing, monitoring of therapeutic effects, and early detection of drug resistance.

## Figures and Tables

**Figure 1 diagnostics-09-00096-f001:**
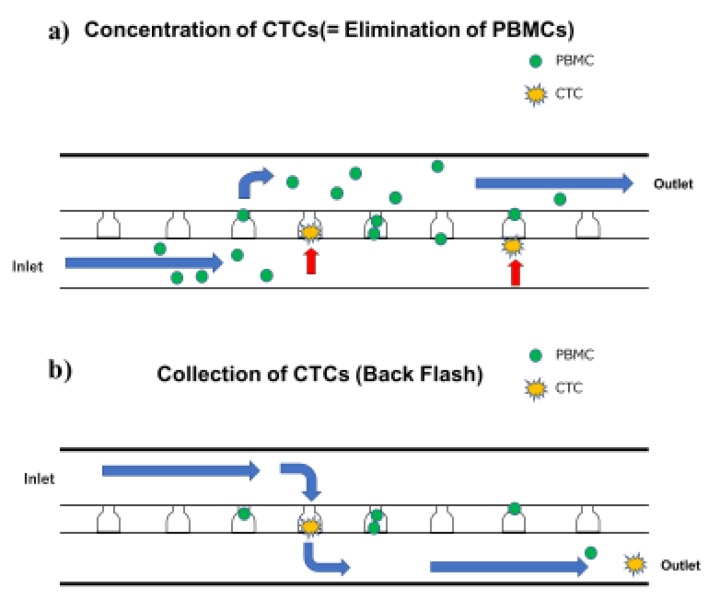
Cell retrieval using Celsee PREP100^®^ combined with On-chip Sort^®^. (**a**) Circulating tumor cells (CTC) enrichment was performed with Celsee. (**b**) CTCs and still contaminating peripheral blood mononuclear cells (PBMCs) were recovered by reverse injection of buffer. Red arrows indicate captured CTC. Blue arrows mean direction of buffer flow.

**Figure 2 diagnostics-09-00096-f002:**
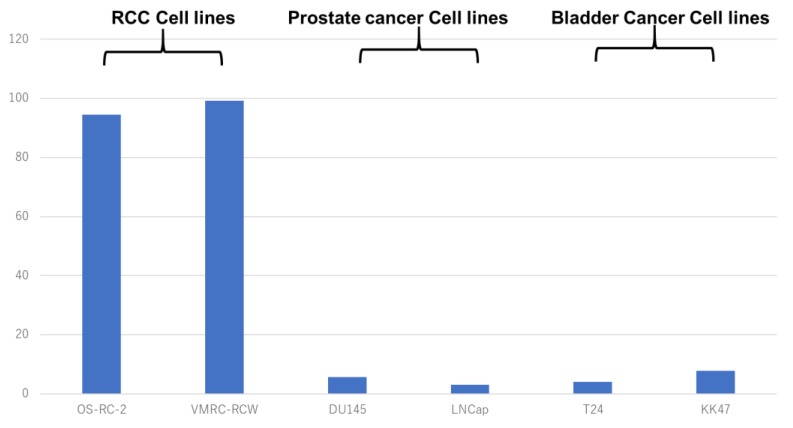
Anti-G250 antibody staining pattern in various cancer cell lines. Flow cytometric analysis of G250 expression on renal cell cancer (RCC) cells and other types of urological cancers (prostate cancers and bladder cancers).

**Figure 3 diagnostics-09-00096-f003:**
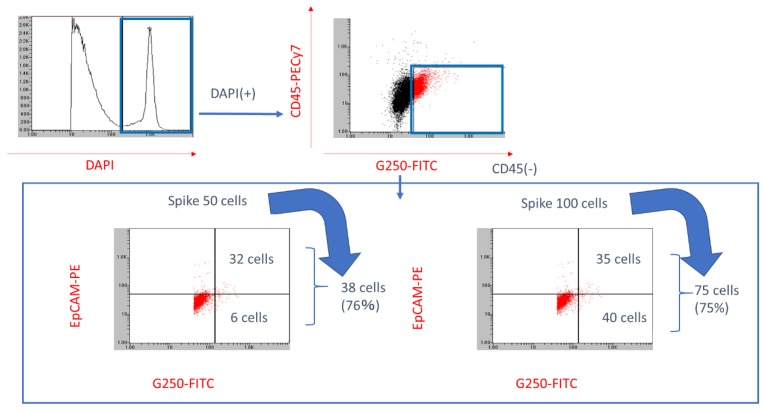
Identification rate of RCC CTC using On-chip Sort^®^ with G250 antibody. Flow cytometric analysis of mixture of PBMCs and VMRC-RCW cells. The cluster of RCC cells is readily identified based on its G250 expression and CD45 negativity. Epithelial cellular adhesion molecule (EpCAM) staining was performed simultaneously as a comparison of G250 staining.

**Figure 4 diagnostics-09-00096-f004:**
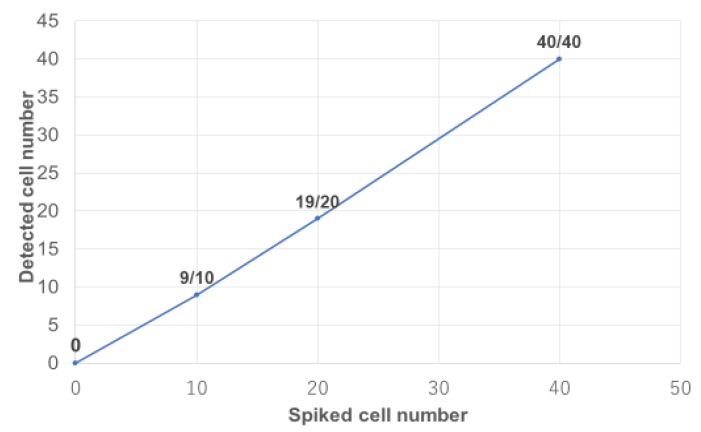
RCC cells concentration using CelSee^®^ and spiked RCC cells counting using On-chip Sort^®^ with anti-G250 antibody. Recovery of known numbers of spiked VMRC-RCW cells whole blood. VMRC-RCW cells (1, 10, 20, 40 cells) were spiked into 4 ml of blood from healthy volunteers. The number of spiked VMRC-RCW cells vs. observed number of recovered cells is plotted.

**Figure 5 diagnostics-09-00096-f005:**
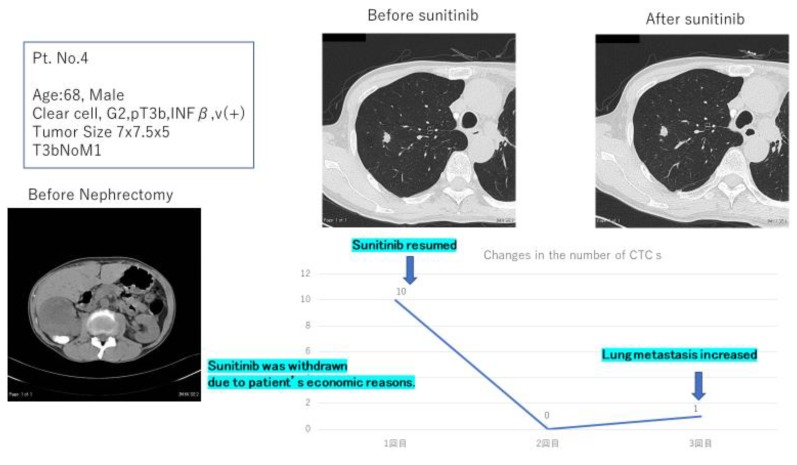
Changes in the number of CTCs during treatment.

**Table 1 diagnostics-09-00096-t001:** Clinicopthologic profiles and detected CTCs in 4 ml of peripheral blood samples from RCC patients.

Pt.No	Sex	Age	Operation	Pathological Result	Tumor Size (cm)	TNM Classification	Number of CTC/4 mL
1	F	65	Total	clear cell, G2, INFα, pT1b	4.5 × 4 × 3	T1bN0M0	1
2	M	68	Partial	papillary, G1, pT1a	1.4 × 0.8	T1aN0M0	2
3	M	57	Total	clear cell, G2, INFβ, pT3a	9 × 6.9	T3aN0M0	3
4	M	68	Total	clear cell, G2, pT3b, INFβ, v (+)	7 × 7.5 × 5	T3bN0M1	10
5	M	61	Total	clear cell, G2>G3, pT3a	3.7 × 5.7	T3aN0M0	1
6	M	70	Nil	unknown	9.5 × 8.1	T3cN0M1	0
7	M	64	Nil	unknown	4.6	T3bN2M1	1
8	M	59	Total	clear cell, G2>G3, pT3a, v (−)	9.5 × 9.5	T1bN0M0	3
9	M	71	Nil	unknown	2.0	T1aN0M0	2
10	M	74	Nil	unknown	17 × 13 × 12	T2bN0M0	0
11	M	74	Total	clear cell, G2, pT1a, v (+)	6.1 × 4.4	T1aN0M0	0
12	M	54	Total	clear cell, G2, pT1a, v (+)	15 × 10	T1aN0M1	3
13	M	58	Nil	unknown	20 × 17 × 15	T2bN0M1	0

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
