# Peer review of "Development of a Highly Sensitive Technique for Capturing Renal Cell Cancer Circulating Tumor Cells"

_diagnostics, 2019, doi:10.3390/diagnostics9030096_

Round 1

Reviewer 1 Report

This excellent work describes an improved method for capturing circulating tumor cells (CTCs) in the blood in renal cell cancer (RCC). The authors used an anti-G250 antibody to detect the CTCs in the blood. The sensitivity in RCC was higher than with established protocols using an antibody targeted against EpCAM. All results are sound and the conclusions drawn are convincing. These results will find the interest of many readers. The topic of liquid biopsies is getting more and more important, and will probably also be used more often in clinical practice. The linguistic style will be sufficient for publication after some improvements. In summary, I strongly support publication of this manuscript. Prior to publication following issue must be solved however:

The abstract is not sufficient and not informative enough. Please summarize more closely your preclinical work as well as your clinical results in the abstract.

Author Response

Dear reviewer.

Thank you for your kind review of my manuscript.

Your opinion is very beneficial to me.

Abstract: purpose: Liquid biopsy is becoming increasingly important as a guide for selecting new drugs and determining their efficacy. In urological cancer, serum markers for renal cell and urothelial cancers has made the development of liquid biopsy for these cancers strongly desirable. Liquid biopsy is less invasive than conventional tissue biopsy is, enabling frequent biopsies and, therefore, is considered effective for monitoring the treatment course. Circulating tumor cells (CTCs) are a representative liquid biopsy specimen. In the present study, we focused on developing our novel technology for capturing renal cell cancer (RCC)-CTCs using an anti-G250 antibody combined with new devices. Basic experiments of our technology showed that it was possible to detect RCC-CTC with a fairly high accuracy of about 95%. Also, RCC-CTC was identified in the peripheral blood of actual RCC patients. Additionally, during the treatment course of the RCC patient, change in the number of RCC-CTC was confirmed in one case. We believe that the technology we developed will be useful for determining the treatment efficacy and drug selection for the treatment of renal cell cancer (RCC). In order to solve issues such as thresholds setting of this technology, large-scale clinical trials are expected.

Thank you.

Michio Naoe M.D.

Reviewer 2 Report

A technique for capturing renal cell cancer (RCC) circulating tumor cells (CTCs) is proposed and tested in this paper. The technique is based on detecting serum markers and is believed less invasive than a conventional tissue biopsy. The technique includes three commercialized products, On-chip sort, CelSee, and G250 antigen. The experiment was designed by putting 0, 10, 20, and 40 VMRC-RCW cells into 4-ml whole blood and the authors tried to capture the cells using the proposed technique. The authors claim that the proposed method can identify RCC CTCs with high accuracy. Overall, the idea is interesting. However, some of the experimental details are not very clear to this reviewer and should be provided. Comments to the authors:

Sec. 2-1, the authors are suggested to provide images, or signal intensities, of anti-G250 stain on the cells. Fig.1 shows cell retrieval with Celsee prep100. What are the pore dimensions of the filter between two channels on the top and the bottom? Why are the dimensions appropriate for the RCC CTCs? While concentrating the CTCs as in Fig.1(a), is the outlet of the bottom channel sealed? If yes, how is it sealed? If no, CTCs may easily flow into the outlet instead of captured by the pores. The resolution of Fig.3 is too low to read. How were the numbers "32 cells", "6 cells", "35 cells", "40 cells" obtained ? The proposed method in FIg.1 can collect not only RCC CTCs but also other cells with similar size. How can the authors be sure the collected cells were all CTCs but not other unknown cells in the whole blood?

Author Response

Dear reviewer.

Thank you for your kind review of my manuscript.

Your opinion is very beneficial to me.

Answer to reviewer 2

Q1: Sec. 2-1, the authors are suggested to provide images, or signal intensities, of anti-G250 stain on the cells.

A1: The results of this flow-cytometric analysis were the percentage of stained cells (cell number), therefore there is no images for this result.

Q2: Fig.1 shows cell retrieval with Celsee prep100. What are the pore dimensions of the filter between two channels on the top and the bottom?

A2: The microfluidic chip method was designed based on the principle that the deformability and diameter of CTCs and peripheral blood mononuclear cells (PBMCs)

Diameter of CTCs and peripheral blood mononuclear cells (PBMCs) differs greatly.

(The diameter of PBMC is about 8 μm, on the other hand,CTC is about 10~20 μm.)

Since the entrance of the tunnel is 25 μm in diameter and it’s exit is 8 μm in diameter, PBMCs can pass through the tunnel, but CTC can not pass through it.

Q3: Why are the dimensions appropriate for the RCC CTCs?

A3: Pore size (Tunnel size: mentioned at Q2) of CelSee is suitable not only for RCC CTC but also other types of CTC.

Q4: While concentrating the CTCs as in Fig.1(a), is the outlet of the bottom channel sealed? If yes, how is it sealed? If no, CTCs may easily flow into the outlet instead of captured by the pores.

A4: Outlet channel is not sealed. However, CTCs do not come out of the outlet before reverse injection. (Switching between forward and reverse injection was performed promptly, the CTC loss is very low.)

Q5: The resolution of Fig.3 is too low to read. How were the numbers "32 cells", "6 cells", "35 cells", "40 cells" obtained?

A5: This study was performed with On-chip Sort. The PBMCs and RCC CTCs were distinguished by staining with anti-D45, anti-G250 antibodies. As a comparison, EpCAM staining was performed simultaneously.

If CTC is defined as CD45(-)/EpCAM(+) cells, some CTCs are missed. However when CTC is defined as CD45(-)/G250(+) cells, approximately 75% of RCC CTCs are detected.

Q6: The proposed method in FIg.1 can collect not only RCC CTCs but also other cells with similar size.

A6: As answered for Q7, by using anti-CD45-Ab combined with anti-G250-Ab, it is possible to distinguish RCC CTC from other cells.

Q7: How can the authors be sure the collected cells were all CTCs but not other unknown cells in the whole blood?

A7: First, red blood cells were destroyed by hemolysis. And the remaining PBMCs and CTCs were distinguished by staining with anti-D45, anti-G250 antibodies. First, the anti-CD45 negative fraction was extracted which means extracted cells were not PBMCs. Then, within anti-CD45 negative cells, stained with G250-Ab was recognized as RCC-CTCs.

Because, G250-Ag is specificically expressed on RCC cells but not for other cells.

Also, at the stage of analysis with OnchipSort, debris and cell fragments are excluded by measuring their size.

Thank you.

Michio Naoe M.D.

Round 2

Reviewer 2 Report

Although the authors have addressed some of this reviewer's concerns, remaining concerns are listed as follows: 

A1: The results of this flow-cytometric analysis were the percentage of stained cells (cell number), therefore there is no images for this result.

In this case, it is important for readers to know the repeatability of the test. (n=?)

A4: Outlet channel is not sealed. However, CTCs do not come out of the outlet before reverse injection. (Switching between forward and reverse injection was performed promptly, the CTC loss is very low.)

Experimental support or theoretical explanation is needed to prove "CTC loss is very low"

A5: ... 

Authors should improve the resolution of figure 3.

A6: As answered for Q7, by using anti-CD45-Ab combined with anti-G250-Ab, it is possible to distinguish RCC CTC from other cells.

If anti-CD45-Ab combined with anti-G250-Ab is enough to distinguish RCC CTC from other cells, the authors need to explain the necessity, or improvement, of using the method in figure 1. 

Author Response

Dear reviewer.

Thank you for your kind review of my manuscript.

Your opinion is very beneficial to me.

Answer to reviewer 2

A1: The results of this flow-cytometric analysis were the percentage of stained cells (cell number), therefore there is no images for this result.

Q to A1: In this case, it is important for readers to know the repeatability of the test. (n=?)

⇒ Actual percentage of gated cells are listed I the graph. Is it OK?   OR It has already been reported that G250 antigen expression is specifically recognized on RCC cells. I mentioned about it in the manuscript (by using references 11~14). Fig2 is just an example of stainability of G250 antibody for urogenital cancers. (This result is not so important.) So, (n) of this analysis is just one. If (n) of it is not sufficient, I would like to omit this figure, and I just would like to mention it in the sentence. (ex: It has been reported that, expression of G250 antigen is specifically recognized on RCC cells. In fact, when we investigated G250 expression on some types of urogenital cancer cell lines (renal cell carcinomas, prostate cancers, bladder cancers), expression of G250 antigen was recognized only in renal cell carcinoma.(data not shown).)  

A4: Outlet channel is not sealed. However, CTCs do not come out of the outlet before reverse injection. (Switching between forward and reverse injection was performed promptly, the CTC loss is very low.)

Q to A4: Experimental support or theoretical explanation is needed to prove "CTC loss is very low"

⇒ The CelSee chip and the collection container are directly connected by a tube. Also, CTC recovery by back-flushing should be performed using a sufficient amount of buffer (8ml) so that no CTC remains in the tube or CelSee chip.

Q to A5: Authors should improve the resolution of figure 3.

⇒ In the Fig3, small and unnecessary characters were deleted.(Only the necessary numerical values are put on the graph. These numbers are very easy to recognize for readers.) Please find attached new Fig3.

A6: As answered for Q7, by using anti-CD45-Ab combined with anti-G250-Ab, it is possible to distinguish RCC CTC from other cells.

Q to A6: If anti-CD45-Ab combined with anti-G250-Ab is enough to distinguish RCC CTC from other cells, the authors need to explain the necessity, or improvement, of using the method in figure 1. 

⇒The sentence below was inserted to the manuscript.

“As a first step, cells other than blood cells are recognized as anti-CD45-Ab negative fractions.Then, within the anti-CD45-Ab negative fractions, cells positive for anti-G250-Ab or anti-EpCAM-Ab are recognized as RCC CTC. However, the sensitivity of these two markers for RCC CTC, sensitivity of anti-G250-Ab is much higher than that of anti-EpCAM-Ab. This is an epoch-making and excellent point compared to the conventional EpCAM based CTC detection.”

Round 3

Reviewer 2 Report

All the concerns of this reviewer have been properly addressed.